# Phosphorylation of Plant Ferredoxin-like Protein Is Required for Intensifying PAMP-Triggered Immunity in *Arabidopsis thaliana*

**DOI:** 10.3390/plants14132044

**Published:** 2025-07-03

**Authors:** Tzu-Yi Chen, Rui-Wen Gong, Bo-Wei Chen, Yi-Hsien Lin

**Affiliations:** Department of Plant Medicine, National Pingtung University of Science and Technology, Pingtung 91201, Taiwan; s082809243g@gmail.com (T.-Y.C.); lo8977sk@gmail.com (R.-W.G.); jwes2115@gmail.com (B.-W.C.)

**Keywords:** agricultural management, bacterial soft rot, casein kinase II, plant ferredoxin-like protein (PFLP), PAMP-triggered immunity (PTI), phosphorylation

## Abstract

The immune response triggered when plant cell surface receptors recognize pathogen-associated molecular patterns (PAMPs) is known as PAMP-triggered immunity (PTI). Several studies have demonstrated that extracellular plant ferredoxin-like protein (PFLP) can enhance PTI signaling, thereby conferring resistance to bacterial diseases in various plants. The C-terminal casein kinase II (CK2) phosphorylation region of PFLP is essential for strengthening PTI. However, whether phosphorylation at this site directly enhances PTI signaling and consequently increases plant disease resistance remains unclear. To investigate this, site-directed mutagenesis was used to generate PFLPT90A, a non-phosphorylatable mutant, and PFLPT90D, a phospho-mimetic mutant, for functional analysis. Based on the experimental results, none of the recombinant proteins were able to enhance the hypersensitive response induced by the HrpN protein or increase resistance to the soft rot pathogen *Pectobacterium carotovorum* subsp. *carotovorum* ECC17. These findings suggest that phosphorylation at the T90 residue might be essential for PFLP-mediated enhancement of plant immune responses, implying that this post-translational modification is likely required for its disease resistance function in planta. To further explore the relationship between PFLP phosphorylation and endogenous CK2, the *Arabidopsis* insertion mutant *cka2* and the complemented line *CKA2R* were analyzed under treatment with flg22_Pst_ from *Pseudomonas syringae* pv. *tomato*. The effects of PFLP on the hypersensitive response, rapid oxidative burst, callose deposition, and susceptibility to soft rot confirmed that CK2 is required for these immune responses. Furthermore, expression analysis of PTI-related genes *FRK1* and *WRKY22/29* in the mitogen-activated protein kinase (MAPK) signaling pathway demonstrated that CK2 is necessary for PFLP to enhance flg22_Pst_-induced immune signaling. Taken together, these findings suggest that PFLP enhances *A. thaliana* resistance to bacterial soft rot primarily by promoting the MAPK signaling pathway triggered by PAMP recognition, with CK2-mediated phosphorylation being essential for its function.

## 1. Introduction

Enhancing plant disease resistance is a vital strategy for protecting crops against pathogen invasion. A key mechanism underlying this defense is the activation of the plant immune system, which is often initiated through the recognition of pathogen-associated molecular patterns (PAMPs) by cell-surface pattern recognition receptors (PRRs). This recognition triggers PAMP-triggered immunity (PTI), which serves as the first line of plant defense [1,2]. PTI activation leads to a suite of immune responses, including the generation of reactive oxygen species (ROS) and callose deposition. In some cases, it can also trigger a hypersensitive response (HR), which together restricts pathogen proliferation [3,4,5].

Numerous studies have demonstrated that enhancing or sustaining PTI responses significantly reduces the incidence and severity of plant diseases [6,7]. Previous research identified a plant ferredoxin-like protein (PFLP) as a positive regulator of PTI signaling. The expression of PFLP in various crop species confers enhanced resistance to bacterial diseases. These include transgenic banana plants resistant to bacterial wilt caused by *Xanthomonas campestris* pv. *musacearum* [8,9,10]; transgenic rice resistant to bacterial blight caused by *Xanthomonas oryzae* pv. *oryzae* [11]; and *Arabidopsis thaliana*, tomato, calla lily, and *Oncidium* orchid resistant to bacterial soft rot [12,13,14,15,16]. In *A. thaliana*, PFLP-mediated resistance has been linked to the amplification of PTI responses, including ROS burst, callose deposition, and a stronger HR, as well as the activation of the mitogen-activated protein kinase (MAPK) pathway [14,17].

PFLP is a ferredoxin protein originally isolated from *Capsicum annuum* [18], and its immune-activating function appears to be closely tied to its structural features [19]. PFLP contains an N-terminal signal peptide, a 2Fe-2S iron–sulfur cluster domain, and a C-terminal casein kinase II phosphorylation (CK2P) site [18]. The 2Fe-2S domain of ferredoxin is classically associated with electron transport in photosynthesis [20], but the PFLP has also been shown to possess antibacterial activity against phytopathogenic bacteria [21]. Although the signal peptide targets PFLP to the chloroplast, transgenic expression of PFLP in the apoplast significantly enhances resistance to bacterial pathogens [22]. Moreover, deletion of the CK2P site in apoplast-localized PFLP abolishes HrpZ-triggered HR and resistance to bacterial wilt, highlighting the functional importance of this phosphorylation site in PFLP-mediated immunity [23].

Plant casein kinase II (CK2) is a highly conserved serine/threonine protein kinase found in all eukaryotes, including plants, where it plays crucial roles in a wide range of physiological processes. CK2 is involved in regulating plant growth and development, light signaling, circadian rhythms, flowering time, and responses to phytohormones such as salicylic acid or abscisic acid. It achieves these functions primarily through the phosphorylation of various substrates, such as transcription factors and regulatory proteins [24,25,26]. Due to its broad range of substrates and regulatory roles, CK2 is considered an essential housekeeping kinase in plant cells. Previous studies indicate that CK2-mediated phosphorylation of PFLP is essential for its immune activity [17,23], but its role in PTI signal amplification and disease resistance remains unclear. To investigate this, we generated PFLP mutants at threonine 90, replacing it with alanine (T90A) or aspartic acid (T90D), and assessed their ability to induce HR in response to HrpN and to confer resistance to *Pectobacterium carotovorum* subsp. *carotovorum*.

In addition, we investigated whether PFLP phosphorylation depends on casein kinase II (CK2) activity by utilizing an *Arabidopsis* CK2 catalytic subunit mutant (*cka2*) and its complementation line (*CKA2R*). We assessed PFLP-mediated immune responses in these genetic backgrounds using the well-established flg22_Pst_-triggered PTI system. Specifically, we evaluated HR, ROS generation, and callose deposition, along with the expression of downstream MAPK target genes *FRK1*, *WRKY22*, and *WRKY29*. We also examined the resistance of these lines to bacterial soft rot. Collectively, this study provides direct evidence that the phosphorylation of PFLP is crucial for the amplification of PTI responses and for enhancing plant resistance to bacterial pathogens.

## 2. Materials and Methods

### 2.1. Plant Materials and Growth Conditions

*Arabidopsis thaliana* ecotype Columbia (Col-0) and the T-DNA insertion mutant *cka2* (casein kinase II, alpha chain 2, AT3G50000, SALK_126662C), obtained from the Arabidopsis Biological Resource Center (ABRC), were used in this study. The *CKA2R* complementation line was generated by amplifying the casein kinase II complementary DNA (cDNA) from Col-0 using the polymerase chain reaction. To clone the Arabidopsis *cka2* gene, cDNA was amplified using CKA2-XbaI-F and CKA2-SacI-R primers (Appendix A). The PCR product was ligated into the pGEM-T vector (Promega, Madison, MI, USA), and the sequence in pGEMT-*cka2* was confirmed. The *cka2* gene fragment was ligated downstream of the CaMV 35S promoter into pBI121 (Takara Bio, Mountain View, CA, USA) and confirmed with 35S/pBI-IndR primers (Appendix A). For *Agrobacterium* transformation, the verified pBI121-*cka2* plasmid was introduced into *Agrobacterium tumefaciens* GV3101. The confirmed *Agrobacterium* GV3101/pBI121-*cka2* strain was used for subsequent plant transformation. The selection of homozygous lines was performed using T_1_ seeds derived from T_0_ plants. Seeds were first germinated on 1/2 Murashige and Skoog (1/2 MS) medium supplemented with 50 μg/mL of kanamycin. Transgenic lines in which more than 95% of seedlings exhibited normal growth and kanamycin resistance were further analyzed via PCR using 35S/PBI-IndR primers to confirm the presence of the transgene in 100% of the plants were regarded as homozygous lines. Homozygous T_2_ plants from these confirmed lines were then used for subsequent experiments. Futhermore, the expression of the *cka2* gene in homozygous lines was confirmed by using At50000-F/ CKA2-SacI-R primers before subsequent experiments (Appendix A). Prior to experiments, seeds were sown in 6-cm-wide pots containing sterilized peat-based soil to allow for germination. One week after sowing, uniformly germinated seedlings were transplanted into new pots. Four-week-old plants were used for all experimental analyses. Plants were grown in a controlled growth chamber under the following conditions: 22 °C, a photoperiod of 16 h light/8 h dark, and a light intensity of 100 μmol m^−2^ s^−1^ (Hipoint, Kaohsiung City, Taiwan).

### 2.2. Preparation of HrpN Protein and flg22_Pst_

The HrpN protein was prepared using the method described in [27]. Briefly, the *Escherichia coli* BL21 harboring pET-HrpN-Ecc17 were cultured in Luria–Bertani (LB) broth at 37 °C for 16 h. The bacterial culture was then inoculated into fresh LB liquid medium at a 1:50 dilution. When the culture reached an OD_600_ of 0.3–0.6, 1 mM isopropyl-β-D-thiogalactopyranoside (IPTG) was added, and further incubation for 16 h was performed to induce protein expression. In all processes, ampicillin at 100 μg/mL was included in all LB broth. The culture was centrifuged at 8000× *g* for 5 min at 4 °C. The pellet was resuspended with 0.1 M phosphate buffer (pH 8.0) and then treated at 100 °C for 10 min after cells were disrupted using an ultrasonic homogenizer (Linko, Mount Gravatt, Australia). The supernatant obtained via centrifugation at 10,000× *g* for 10 min was used as the HrpN protein for subsequent experiments. To confirm the HrpN protein, 1.0 μg of protein extract was fractionated in 15% sodium dodecyl sulfate-polyacrylamide gel electrophoresis and visualized at around 40 kDa. The stock solution of flg22_Pst_ (TRLSSGLKINSAKDDAAGLQIA) [28] was prepared from the commercially synthesized peptide purchased from LifeTien LCC (South Plainfield, NJ, USA) with 25 mM Tris-HCl buffer (pH 7.5). The stock solution was used in all experiments at a final concentration of 0.5 μM in this study.

### 2.3. Preparation of Recombinant PFLP Proteins

To investigate whether the presence of the CK2 phosphorylation (CK2P) site in PFLP affects plant disease resistance, comparative analyses were conducted using three types of recombinant PFLP proteins: the full-length PFLP protein, PEC (a variant lacking the AELVG amino acid sequence but retaining the CK2P site), and PDC (a variant lacking the CK2P site), as described in our previous study [23]. To further explore whether the phosphorylation of PFLP is associated with enhanced plant immunity, site-directed mutagenesis was used to substitute the threonine (T) at position 90 with either alanine (A), which cannot be phosphorylated, or aspartic acid (D), which mimics a phosphorylated residue synthesized de novo. Recombinant proteins carrying these point mutations were expressed using *Escherichia coli* expression systems with synthetic plasmids pET16b-PFLPT90A and pET16b-PFLPT90D (Genewiz, NJ, USA). These plasmids were transformed into *E. coli* BL21 cells for protein expression. For protein purification, bacterial cultures were first centrifuged at 8000× *g* for 5 min at 4 °C, and the pellets were resuspended in lysis buffer (50 mM Tris-HCl, 300 mM NaCl, and 10 mM imidazole, pH 8.0). Cells were disrupted using an ultrasonic homogenizer, followed by centrifugation at 10,000× *g* for 10 min. The resulting supernatants were subjected to purification using a Ni-NTA Spin Kit (Qiagen, Hilden, Germany), and the recombinant proteins were eluted with elution buffer (50 mM Tris-HCl, 300 mM NaCl, and 250 mM imidazole, pH 8.0). The purified proteins were then concentrated using Amicon^®^ Ultra centrifugal filters with a 10 kDa molecular weight cutoff (Merck, Taufkirchen, Germany). The purified recombinant protein was quantified (1 μg) and analyzed via protein gel electrophoresis (SDS-PAGE). After gel separation, the recombinant protein was stained with commassie blue and then blotted with Immun-Blot^®^ PVDF Membrane (Bio-Rad, Hercules, CA, USA). The blotted PVDF Membrane was immersed in anti-His antibody (1:3000, Rockland, UK) at room temperature for 16–18 h, then reacted with HRP-conjugated goat anti-rabbit immunoglobulin G (IgG) (1:3000, Rockland, UK) at room temperature for 1 h. After washing, the PVDF membrane was stained with Metal Enhanced DAB Substrate Kit (Thermo, Waltham, MA, USA) to confirm the expression of the recombinant protein. The confirmed protein solutions (Appendix A) were used for subsequent experiments.

### 2.4. Analysis of Hypersensitive Response Ratio in Arabidopsis thaliana

To evaluate the effect of recombinant PFLP proteins on the hypersensitive response (HR) induced by HrpN, a previously established method was followed using *A. thaliana* Col-0 [23]. Purified recombinant proteins and the HrpN protein were prepared in mixtures at a final concentration of 0.5 mg/mL and 0.5mg/mL, respectively. The control treatment consisted of 25 mM Tris-HCl buffer (pH 7.5). For the infiltration of each plant, three mature leaves were selected, and small puncture wounds were made on both sides of each leaf using a needle. The prepared mixtures were then infiltrated into the wounded sites, with each treatment applied to more than 30 plants per experiment. Each experiment was independently repeated three times. After infiltration, plants were incubated in a growth chamber. HR symptoms were evaluated 24 h post-treatment. The HR ratio was calculated using the formula: HR (%) = (Nn/6) × 100, where Nn represents the number of sites exhibiting necrosis, and 6 is the total number of infiltration sites per plant. Statistical analysis was performed using Tukey’s HSD test at a 5% significance level.

### 2.5. Rapid Reactive Oxygen Species Generation and Callose Deposition Assay

To evaluate whether reactive oxygen species (ROS) generation and callose deposition were enhanced by PFLP proteins during the activation of PAMP-triggered immunity (PTI), flg22_Pst_ was co-infiltrated with recombinant proteins into *Arabidopsis* leaves. The final concentrations of each PFLP protein and flg22_Pst_ in the mixture were 0.5 mg/mL and 0.5 μM, respectively. For the ROS generation assay, infiltrated leaf strips were stained and examined via fluorescence microscopy at 1 h post-infiltration [14]. Leaves were stained with 20 μM 2ʹ,7ʹ-dichlorodihydrofluorescein diacetate (Molecular Probes, OR, USA) and observed under a fluorescence microscope equipped with a filter set for excitation at 465–495 nm and emission at 515–555 nm (Leica Microsystems, Wetzlar, Germany). For the callose deposition assay, leaf strips were observed at 8 h post-infiltration. Prior to staining, the samples were decolorized by soaking in 95% ethanol overnight, followed by staining with 0.01% aniline blue (Sigma, St. Louis, MO, USA) for 2 h [17]. Fluorescence was then observed using a filter set with excitation at 340–380 nm and emission at 400–425 nm. The intensity of ROS generation and callose deposition was quantified using ImageJ software version 1.54j (https://imagej.net/ij/, accessed on 1 July 2024). Data were collected from 30 individual leaf samples per treatment. Statistical analysis was performed using Tukey’s HSD test at a 5% significance level.

### 2.6. Disease Severity Assay

To evaluate the ability of PFLP proteins to suppress bacterial soft rot in *Arabidopsis* plants, a disease severity assay was conducted using a detached leaf inoculation method [14]. Briefly, bacterial suspensions of *Pectobacterium carotovorum* subsp. *carotovorum* strain Ecc17 [17] from cabbage were prepared in sterile distilled water and adjusted to an OD_600_ of 0.3 (around 1.0 × 10^8^ CFU/mL). Detached *Arabidopsis* leaves were immersed and vacuum-infiltrated with a 100-fold dilution of the bacterial suspension at 200 mm Hg for 15 min. The inoculated leaves were then incubated in a growth chamber at 22 °C. Disease severity was assessed at 48 h post-inoculation using a rating scale from 0 to 4 based on the proportion of leaf area exhibiting soft rot symptoms (0 = no symptoms; 1 = 0–25% soft rot area; 2 = 25–50%; 3 = 50–75%; 4 = 75–100%). For each treatment, disease severity data were collected from five leaves, and each treatment was replicated three times for statistical analysis.

### 2.7. Analysis of Defense Gene Expression Involved in the MAPK Signaling Pathway

To investigate whether casein kinase II is involved in the enhancement of defense gene expression by PFLP during flg22_Pst_-triggered immune responses, the expression levels of marker genes associated with the intracellular MAPK signaling pathway were analyzed. Four-week-old leaves of *Arabidopsis thaliana* Col-0 wild-type and the *cka2* mutant were infiltrated with either flg22_Pst_ alone or a mixture containing flg22_Pst_ (0.5 μM) and PFLP recombinant protein (0.5 mg/mL). Leaf samples were collected at 0.5 and 24 h post-treatment. Total RNA was extracted from 50 mg of plant tissue using the Total RNA Mini Kit (Geneaid, New Taipei City, Taiwan) according to the manufacturer’s protocol. RNA purity was measured, and the Transcriptor First Strand cDNA Synthesis Kit (Roche, Basel, Switzerland) was further used to synthesize complementary DNA (cDNA) via reverse transcription. Then, 200 ng of cDNA was used in subsequent real-time quantitative PCR (qPCR) analysis of MAPK pathway marker genes. qPCR was performed using the MiniOpticon™ system (Bio-Rad, Hercules, CA, USA). Each 10 μL reaction contained 1 × iQ™ SYBR Green Supermix (Thermo Fisher Scientific, Waltham, MA, USA), 500 nM gene-specific forward and reverse primers (see Appendix A) [29], and 200 ng of cDNA template. The tubulin gene was used as an internal reference [30]. The thermal cycling conditions were as follows: Step 1, 95 °C for 3 min; Step 2, 35 cycles of 95 °C for 10 secs and 50 °C for 30 secs. The expression of *TUB2* was used as a reference gene, and the relative fold induction was normalized via uninfiltrated treatment [31,32]. After completion of the qPCR amplification cycles, melting curve analysis was performed to verify the specificity of the PCR products. Each treatment was analyzed with five replicates in this assay.

## 3. Results

### 3.1. Effect of PFLP Recombinant Protein on the Occurrence of Hypersensitive Reaction (HR) and Its Effect on Controlling Bacterial Soft Rot Disease

Since the occurrence of HR is an important indicator of plant disease resistance, to confirm whether the CK2P site of PFLP would affect the disease resistance of plants, the effect of two versions of truncated PFLP, PEC (extant CK2P site), and PDC (deleted CK2P site) on HR triggered by harpin (HrpN) derived from *Pectobacterium carotovorum* subsp. *carotovorum* was assessed. The results showed that only PFLP and PEC recombinant proteins enhanced the HrpN-induced hypersensitive response in Arabidopsis Col-0, while PDC had no enhancing effect at all (Figure 1A). Further analysis of HrpN-induced HR using recombinant proteins with point mutations in the CK2P site of PFLP showed that non-phosphorylated PFLPT90A was indeed unable to enhance HrpN-induced HR. However, mimicking phosphorylation of PFLPT90D also failed to enhance the HR ratios induced by HrpN (Figure 1B). In terms of the results of the disease control performed with different recombinant proteins against bacterial soft rot, it was shown that PFLP recombinant protein treatment could reduce the disease severity of leaves, and most leaves showed mild or no soft rot symptoms when compared to the treatment of Ecc17 alone (Figure 1C,D). However, neither the treatment with PFLPT90A nor PFLPT90D recombinant protein could reduce the occurrence of the disease, and most leaves showed severe soft rot symptoms.

### 3.2. Effect of PFLP Recombinant Protein on Enhancing PTI Defense Response in the cka2 Mutant

To investigate whether the phosphorylation of PFLP by casein kinase II in plants is indeed involved in enhancing PTI defense signaling, flg22_Pst_ was used as a PAMP to analyze the effects of PFLP recombinant protein on defense responses, such as the rapid generation of reactive oxygen species (ROS) and callose deposition, in the *cka2* mutant.

The results showed that treatment with flg22_Pst_ in *A. thaliana* Col-0 plants induced rapid ROS generation, and this response was further enhanced by the addition of PFLP recombinant protein (Figure 2A). The quantification of fluorescence intensity revealed that PFLP significantly increased flg22_Pst_-induced fluorescence by 47.4-fold (Figure 2B). However, in the *cka2* mutant, PFLP was unable to enhance flg22_Pst_-induced ROS generation; its fluorescence intensity was not significantly different from that of flg22_Pst_ treatment alone (Figure 2C).

Similarly, flg22_Pst_ induced a small amount of callose deposition in Col-0 plants (Figure 2D), and this response was significantly enhanced by PFLP, which increased flg22_Pst_-induced callose fluorescence by 10.1-fold (Figure 2E). However, the enhancement of callose deposition by PFLP was abolished in the *cka2* mutant, with fluorescence levels not significantly different from flg22_Pst_ treatment alone (Figure 2F).

### 3.3. Disease Control of PFLP Recombinant Protein in cka2 Mutant

To confirm whether the role of PFLP in enhancing disease resistance to soft rot is related to its phosphorylation by casein kinase II in Arabidopsis, the PFLP recombinant protein was used to carry out a test on the control of bacterial soft rot in the *cka2* mutant. The results exhibited that the PFLP protein showed superior control results in terms of reducing the disease severity and soft rot symptoms in Col-0 plants (Figure 3A,B). However, the occurrence of soft rot could not be reduced in the *cka2* mutant. Most leaves showed severe soft rot symptoms in both the treatments of PFLP and Tris buffer (control) in the *cka2* mutant; the disease severities were over 79.8% in both treatments without significant difference (Figure 3C,D).

### 3.4. Relationship Between PFLP-Mediated Enhancement of Disease Resistance and MAPK Pathway Signaling in the cka2 Mutant

PFLP may intensify PTI responses and confer resistance against soft rot disease by being phosphorylated by CK2. To further investigate whether the *A. thaliana cka2* mutant affects the MAPK signaling pathway during the PFLP-mediated intensification of PTI, the expression of marker genes *FRK1*, *WRKY22*, and *WRKY29* in the MAPK pathway was analyzed. For FRK1 expression, no significant differences were observed between Col-0 and the *cka2* mutant at 0.5 h post-infiltration with either flg22_Pst_ alone or flg22_Pst_ combined with PFLP, compared to the Tris control (Figure 4(A-1,B-1)). In Col-0 plants, flg22_Pst_ treatment for 24 h led to a 10.6-fold induction of *FRK1* expression relative to the Tris buffer control. Notably, co-treatment with PFLP resulted in a 46.6-fold induction, significantly greater than flg22_Pst_ alone. (Figure 4(C-1)). In contrast, although flg22_Pst_ treatment in the *cka2* mutant induced *FRK1* expression by 15.6-fold, co-treatment with PFLP only induced a 23.4-fold increase, with no significant difference between the two treatments (Figure 4(D-1)). Regarding *WRKY22* expression, in the Col-0 plant at 0.5 h post-infiltration, flg22_Pst_ alone induced a 1.6-fold increase compared to the Tris control, while co-treatment with PFLP led to a 2.9-fold increase, which remained significantly higher than flg22_Pst_ alone at 24 h (Figure 4(A-2,C-2)). In the *cka2* mutant, flg22_Pst_ treatment induced a 2.6-fold increase at 0.5 h; however, co-treatment with PFLP failed to further enhance *WRKY22* expression, a trend that persisted at 24 h (Figure 4(B-2,D-2)). For *WRKY29* expression, both Col-0 and *cka2* plants exhibited patterns similar to *FRK1*. In the Col-0 plant at 0.5 h, neither flg22_Pst_ alone nor its combination with PFLP resulted in significant changes compared to the Tris control (Figure 4 (A-3,B-3)). At 24 h, flg22_Pst_ and its combination with PFLP induced 1.5-fold and 3.0-fold increases in Col-0, respectively (Figure 4(C3)). In the *cka2* mutant, the same treatments induced 2.3-fold and 3.2-fold increases, respectively, both significantly higher than the Tris control but not significantly different from each other (Figure 4(D-3)).

### 3.5. Effects of PFLP on flg22_Pst_-Induced Reactive Oxygen Species Generation and Callose Deposition in the CKA2R Complementation Line

To confirm that CK2 is indeed required for the PFLP-mediated intensification of the flg22_Pst_-induced PTI pathway, a complementation line (*CKA2R*), in which the *cka2* mutant overexpresses the *cka2* gene, was used to analyze reactive oxygen species (ROS) generation and callose deposition. In terms of ROS generation, recombinant PFLP protein failed to enhance flg22_Pst_-induced ROS generation in the *cka2* mutant, but significantly increased ROS generation in the *CKA2R* line (Figure 5A). Quantitative analysis revealed that, compared to the Tris buffer control, flg22_Pst_ treatment in *CKA2R* induced a 3.2-fold increase in relative fluorescence, whereas co-treatment with PFLP resulted in a 198.8-fold increase, which was significantly higher than flg22_Pst_ alone (Figure 5B). When analyzing different *Arabidopsis* plants under co-infiltration with flg22_Pst_ and PFLP, the results showed that the *cka2* mutant failed to induce rapid ROS generation. In contrast, both Col-0 and the *CKA2R* exhibited significant ROS generation, with no significant difference between them (Figure 5C). Similarly, callose deposition was not intensified by PFLP in the *cka2* mutant but was markedly increased in the *CKA2R* line (Figure 5D). Quantification based on the Tris control showed that flg22_Pst_ treatment alone induced a 465.6-fold increase in fluorescence intensity in *CKA2R*, while co-treatment with PFLP led to a dramatic 3184.2-fold increase, significantly greater than flg22_Pst_ alone (Figure 5E). Upon co-infiltration with flg22_Pst_ and PFLP, analysis of various *Arabidopsis* plants revealed that the *cka2* mutant triggered only a slight signal of callose deposition. In contrast, both Col-0 and the *CKA2R* showed substantial callose deposition, without significant difference (Figure 5F).

### 3.6. Assay of PFLP-Mediated Control of Bacterial Soft Rot in the CKA2R Complementation Line

To further confirm that CK2 is also required for PFLP-mediated enhancement of resistance to bacterial soft rot in *Arabidopsis*, the *CKA2R* complementation line was used to evaluate the effect of PFLP treatment on disease suppression. The results showed that the protective effect of PFLP against soft rot disease was completely abolished in the *cka2* mutant (Figure 6A). In contrast, this protective effect was restored in *CKA2R*, and significant disease suppression was observed (Figure 6B). Similarly, wild-type Col-0 plants treated with PFLP exhibited a marked reduction in disease severity and soft rot symptoms (Figure 6C,D).

## 4. Discussion

Numerous studies have demonstrated that the plant ferredoxin-like protein (PFLP) derived from sweet pepper can enhance resistance to various bacterial pathogens in different plant species by intensifying plant innate immunity [9,10,11,12,13,15,16,22]. Subsequent research has confirmed that the primary mechanism by which PFLP enhances disease resistance is through activation of the MAPK pathway during the PAMP-triggered immunity (PTI) response [14,17]. Moreover, previous findings have shown that the casein kinase II phosphorylation site (CK2P site) located at the C-terminal region of PFLP is required for its disease resistance-enhancing function [23]. However, it remains unclear whether actual phosphorylation at this CK2P site is essential for PFLP’s role in promoting PTI signaling.

In this study, we first employed site-directed mutagenesis to investigate whether phosphorylation at the CK2P site is necessary for PFLP-mediated disease resistance. Using the hypersensitive response (HR) induced by HrpN, a harpin protein from *Pectobacterium carotovorum* subsp. *carotovorum*, we demonstrated that the presence of the CK2P site is indeed crucial. Both the non-phosphorylatable mutant PFLPT90A and the phosphomimic mutant PFLPT90D failed to enhance HrpN-induced HR or confer resistance to bacterial soft rot. This finding is similar to Ataxin-1, a protein associated with spinocerebellar ataxia type 1, where phosphomimic substitutions could not fully replicate the effect of actual phosphorylation [33]. In the study of phototropism protein NPH3 in *A. thaliana*, the results exhibit that both the non-phosphorylatable (S744A) and the phosphomimetic (S744D/S744E) mutants of NPH3 fail to interact with 14-3-3 proteins, highlighting the requirement for authentic phosphorylation at serine 744 [34]. Therefore, our results suggest that actual phosphorylation of PFLP is required in planta for disease resistance enhancement.

To further support the role of phosphorylation in PFLP function, we investigated the involvement of host kinases. A study showed that the CK2P site in the C-terminal region of the plant TSPY protein can be phosphorylated by extracellular casein kinase II (CK2) [35]. This led us to hypothesize that the CK2P site in PFLP might also be phosphorylated by plant CK2 to enable PFLP to more effectively amplify the PTI defense responses. Analysis using the *cka2* mutant, which lacks the CK2 catalytic subunit encoded by At3g50000, revealed that PFLP failed to enhance HrpN-induced HR in this background. This indicates that CK2 is required for PFLP to promote PAMP-induced hypersensitive responses. To further verify the importance of CK2 in PFLP-mediated enhancement of PTI signaling, we used flg22_Pst_, a well-characterized PAMP, as the elicitor. The *cka2* mutant itself showed normal ROS production and callose deposition upon flg22_Pst_ treatment. However, the ability of PFLP to further enhance these PTI responses was completely abolished in the *cka2* mutant. These results indicate that CK2-mediated phosphorylation of PFLP might be necessary for intensifying PTI signaling outputs.

Consistently, PFLP also failed to reduce soft rot symptoms in *cka2* mutants, suggesting that CK2 is essential for PFLP to confer resistance against *P. carotovorum* subsp. *carotovorum*. Since the PFLP-induced enhancement of disease resistance is known to be linked with activation of the MAPK signaling cascade [17], we next examined whether CK2 also affects PFLP’s ability to activate MAPK-related defense genes [36,37,38]. Using *FRK1*, a well-known PTI marker gene regulated by the MAPK cascade [39], we observed that PFLP-induced upregulation of *FRK1* expression was abolished in *cka2* mutants. Additionally, the expression of *WRKY22* and *WRKY29*, two downstream transcription factors activated by MPK3/6 [36], was significantly enhanced by PFLP in wild-type plants but not in *cka2* mutants. These findings indicate that CK2 plays a critical role in the PFLP-mediated enhancement of MAPK pathway activation during PTI, which is essential for resistance to soft rot disease.

To further validate this conclusion, a CK2-complemented line (*CKA2R*), generated by overexpressing the cka2 gene via *Agrobacterium*-mediated transformation in the *cka2* mutant background, was analyzed. When challenged with flg22_Pst_, this complemented line restored the ability of PFLP to enhance ROS generation, callose deposition, and resistance to soft rot. However, numerous studies have demonstrated that CK2 also functions as an extracellular kinase, broadly influencing the signaling of various plant hormones, including salicylic acid (SA) and abscisic acid (ABA) [24,25,26]. We cannot rule out the possibility that the *cka2* mutant may affect defense responses through alternative pathways, and we currently do not have direct evidence showing that PFLP is indeed phosphorylated by CK2. Nevertheless, it is reasonable to infer that the presence of CK2 is associated with the enhanced PTI responses mediated by PFLP. When considered alongside the observed increase in plant immune responses to soft rot disease, these results suggest that CK2-mediated phosphorylation of PFLP likely contributes to the amplification of PTI, thereby strengthening plant immunity. In addition, CK2 may also participate in the regulation of other defense-related pathways.

Moreover, in this study, the use of exogenous recombinant proteins provided a tractable and timely system to dissect the functional contributions of PFLP variants. This strategy avoided the potential complications associated with stable transformation in mutant backgrounds and allowed us to evaluate the plant immune-enhancing activity of each protein variant under standardized conditions. Our observations of ROS generation and callose deposition following treatment with recombinant PFLP proteins support their biological activity in planta. Nevertheless, future studies incorporating transgenic lines will be essential to confirm the *in planta* relevance of our findings and to further elucidate the role of CK2-mediated phosphorylation in PFLP-dependent immune responses.

Taken together, our findings demonstrate that PFLP enhances *Arabidopsis* resistance to bacterial soft rot primarily by amplifying PAMP-triggered immunity, including HR, rapid ROS production, and callose deposition. These defense responses might depend on CK2-mediated phosphorylation of PFLP, which promotes activation of the MAPK signaling cascade during PTI. Thus, CK2 is a critical regulator required for PFLP to potentiate PTI and confer effective disease resistance in plants.

## Figures and Tables

**Figure 1 plants-14-02044-f001:**
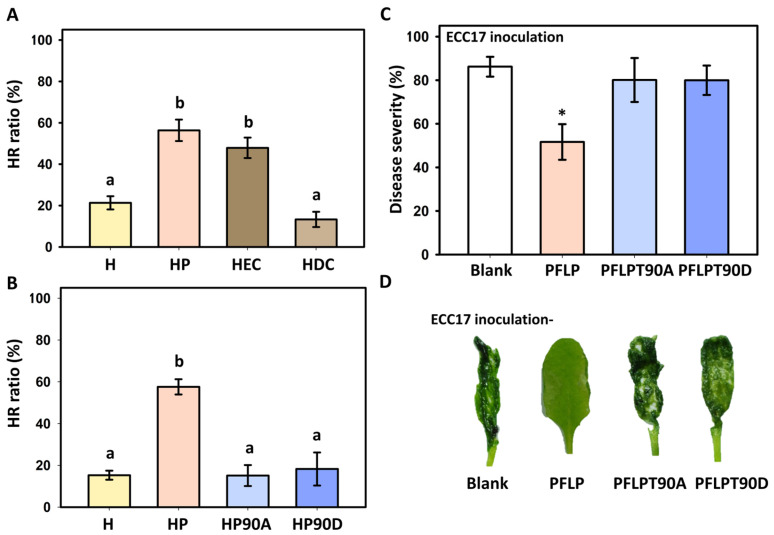
Effects of recombinant PFLP proteins on hypersensitive response (HR) ratio and disease control of bacterial soft rot in *Arabidopsis thaliana* Col-0. (**A**) HR intensification by different truncated PFLP mutants. In the symbols below the figure, H indicates the treatment with HrpN, while P, EC, and DC indicate the treatments containing PFLP and its two truncated mutants, PEC and PDC, respectively. (**B**) HR induction by single amino acid substitutions in PFLP proteins upon HrpN induction. The symbol H indicates the treatment with HrpN, while P, P90A, and P90D indicate the treatments containing PFLP and its two single amino acid substituted mutants, PFPLT90A and PFLPT90D, respectively. HR analysis was conducted using mixtures of HrpN and PFLP derivatives at 0.5 mg/mL, and HR ratios in Arabidopsis plants were recorded 24 h post-infiltration. (**C**) The disease control effects of PFLP proteins against bacterial soft rot at 24 h post-infiltration. (**D**) Symptom appearance of bacterial soft rot. Inoculation was carried out by infiltrating mixtures of *Pectobacterium carotovorum* subsp. *carotovorum* Ecc17 and recombinant PFLP proteins. Different letters above columns indicate significant differences based on Tukey’s HSD test (*p* < 0.05). An asterisk indicates significant differences compared to inoculation with Ecc17 alone (Blank) based on a *t*-test (*p* < 0.05).

**Figure 2 plants-14-02044-f002:**
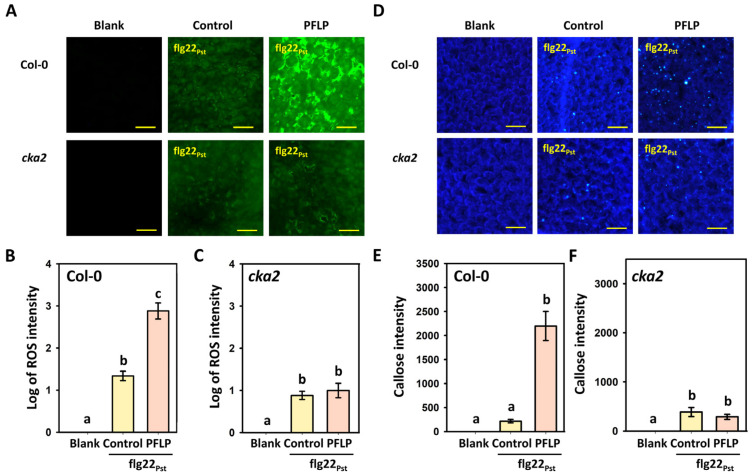
Effects of recombinant PFLP proteins on flg22*_Pst_*-induced reactive oxygen species (ROS) generation and callose deposition in *Arabidopsis thaliana* plants. Leaves of four-week-old plants were infiltrated with mixtures of PFLP and flg22*_Pst_* at 0.5 mg/mL and 0.5 μM, respectively. (**A**) Images of ROS generation, with yellow bars indicating 80 μm in length. (**B**,**C**) The fluorescence intensity of ROS generation in Col-0 and *cka2* plants, respectively. (**D**) Images of callose deposition, with yellow bars indicating 80 μm in length. (**E**,**F**) The fluorescence intensity of callose deposition in Col-0 and *cka2* plants, respectively. At least thirty infiltrated leaf samples of each treatment were analyzed as replicates in this assay. Different letters indicated significant differences based on Tukey’s HSD test (*p* < 0.05).

**Figure 3 plants-14-02044-f003:**
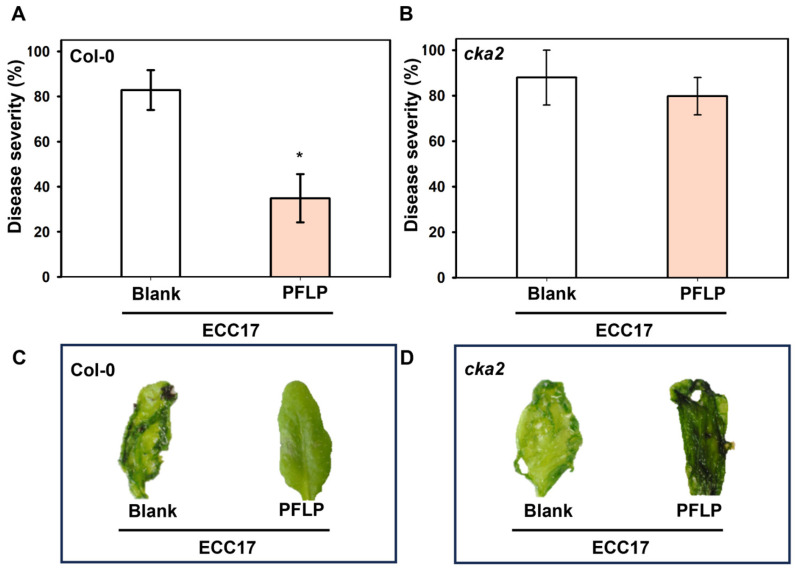
Effect of recombinant PFLP proteins on disease control of bacterial soft rot in the *Arabidopsis thaliana* plants. (**A**,**B**) The disease control of PFLP on Col-0 and *cka2* plants, respectively. Inoculation was performed on leaves of four-week-old plants by infiltrating the mixtures of *Pectobacterium carotovorum* subsp. *carotovorum* Ecc17 and PFLP. (**C**,**D**) The soft rot symptoms on Col-0 and *cka2* plants, respectively. The evaluation was performed at 24 h post-infiltration. In each treatment, results from nine infiltrated leaves were calculated as one repeat, and each treatment had three repeats in one assay. An asterisk indicates significant differences compared to the inoculation performed with Ecc17 alone (Blank) based on *t*-test (*p* < 0.05).

**Figure 4 plants-14-02044-f004:**
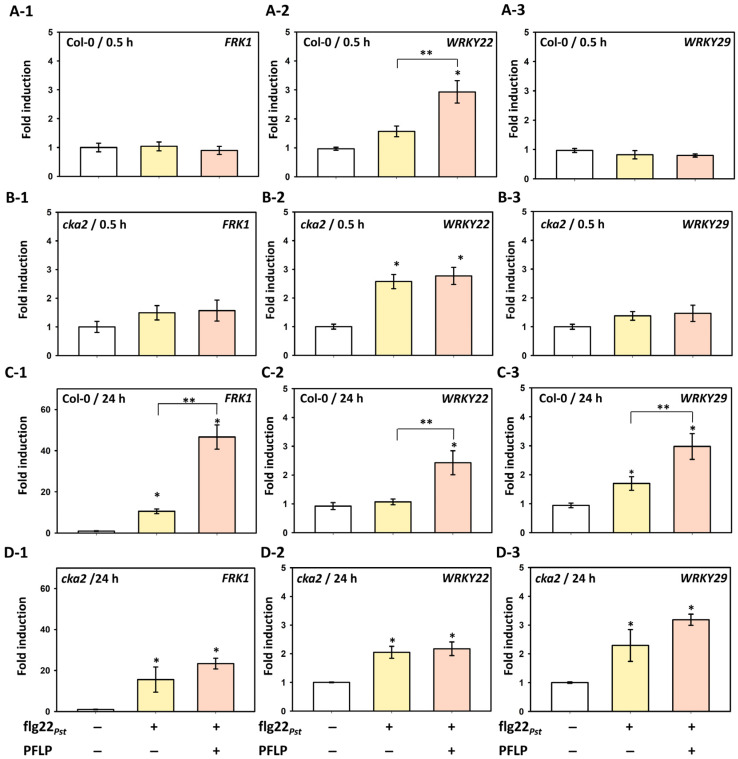
Effects of PFLP protein on the expression of *FRK1*, *WRKY22* and *WRKY29* transcripts in *Arabidopisis thaliana* plants upon flg22*_Pst_* treatment. ((**A-1**)–(**A-3**)) The relative gene expression of *FRK1*, *WRKY22,* and *WRKY29*, respectively, in Col-0 plants at 0.5 h post-infiltration. ((**B-1**)–(**B-3**)) The relative gene expression in *cka2* plants at 0.5 h post-infiltration. ((**C-1**)–(**C-3**)) The relative gene expression in Col-0 plants at 24 h post-infiltration. ((**D-1**)–(**D-3**)) The relative gene expression in *cka2* plants at 24 h post-infiltration. Leaves of four-week-old plants were infiltrated with mixtures of PFLP and flg22*_Pst_*. The symbols “+” and “−” indicate the inclusion or exclusion of treatments with flg22*_Pst_* and PFLP, respectively. Fold induction of each treatment was normalized by blank treatment at 0 h. An asterisk indicates significant differences compared to blank treatment based on *t*-test (*p* < 0.05). Double asterisks indicate significant differences compared to flg22*_Pst_* alone treatment based on *t*-test (*p* < 0.05).

**Figure 5 plants-14-02044-f005:**
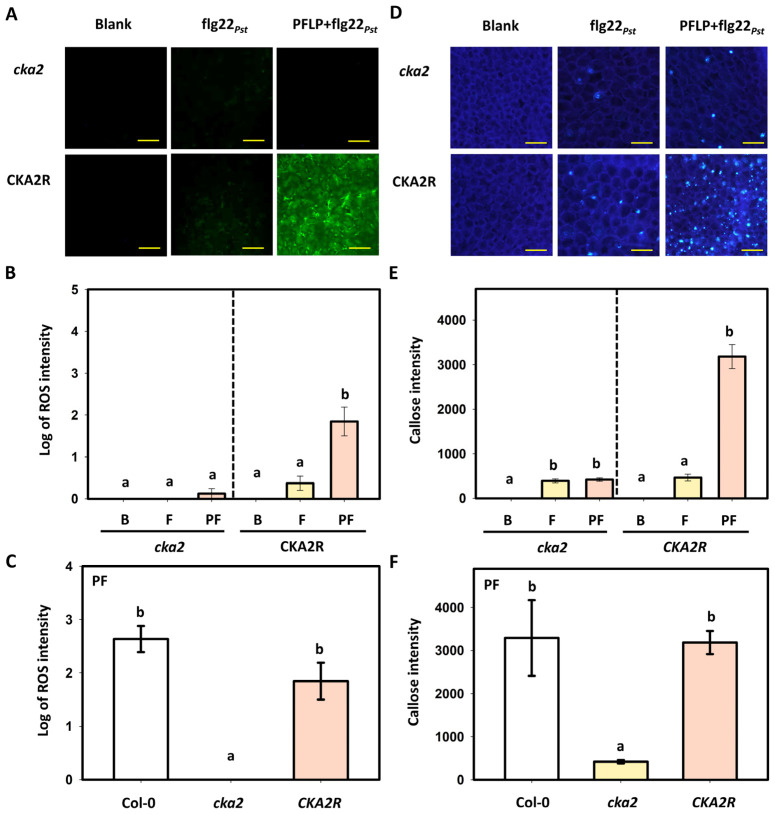
Effects of PFLP protein on flg22*_Pst_* induced reactive oxygen species (ROS) generation and callose deposition in the *Arabidopsis thaliana CKA2R* complementation line. Leaves of four-week-old plants were infiltrated with mixtures of PFLP and flg22*_Pst_*. (**A**) Images of ROS generation in *cka2* and *CKA2R* plants, with yellow bars indicating 80 μm in length. (**B**) The fluorescence intensity of ROS in these plants. B indicates the blank treatment with Tris buffer control, F indicates the treatment with flg22_Pst_ alone, and PF indicates the co-treatment with flg22_Pst_ and PFLP. (**C**) The generation of ROS in different *Arabidopsis* plants under co-treatment with flg22_Pst_ and PFLP. (**D**) The images of callose deposition in *cka2* and *CKA2R* plants, with yellow bars indicating 80 μm in length. (**E**) The fluorescence intensity of callose in these plants. (**F**) Callose deposition in different *Arabidopsis* plants under co-treatment with flg22_Pst_ and PFLP. At least thirty infiltrated leaf samples of each treatment were calculated as repeats in this assay. Different letters indicated significant differences based on Tukey’s HSD test (*p* < 0.05).

**Figure 6 plants-14-02044-f006:**
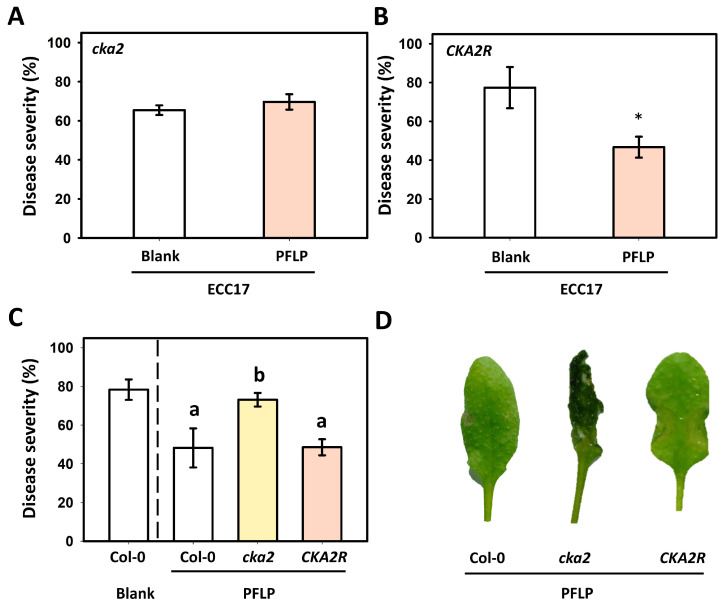
Effect of PFLP-mediated disease control on bacterial soft rot in the *Arabidopsis thaliana CKA2R* complementation line. (**A**,**B**) The disease severity in *cka2* mutant and *CKA2R* plants, respectively. Inoculation was performed on leaves of four-week-old plants by infiltrating the mixtures of *Pectobacterium carotovorum* subsp. *carotovorum* Ecc17 and PFLP recombinant proteins. (**C**) The differences in disease severity of bacterial soft rot among distinct *Arabidopsis* plants following PFLP treatment. (**D**) Soft rot symptoms on Col-0, *cka2* mutant, and *CKA2R* plants, respectively, upon PFLP treatment. The disease severity and symptoms of infiltrated leaves were observed at 24 h post-infiltration. In each treatment, nine infiltrated leaves were used to calculate the disease severity as one repeat, and each treatment had three repeats. An asterisk indicates significant differences compared to the inoculation performed with Ecc17 alone (Blank) based on *t*-test (*p* < 0.05). Different letters indicate significant differences based on Tukey’s HSD test (*p* < 0.05).

## Data Availability

The original contributions presented in the study are included in the article, further inquiries can be directed to the corresponding author.

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
