# Peer review of "Phosphorylation of Plant Ferredoxin-like Protein Is Required for Intensifying PAMP-Triggered Immunity in Arabidopsis thaliana"

_plants, 2025, doi:10.3390/plants14132044_

Round 1

Reviewer 1 Report

Comments and Suggestions for Authors

The authors have made some interesting findings regarding the mechanism by which PFLP enhances PTI responses, showing that the key phosphorylation site is necessary for these responses but that in vivo phosphorylation is required. Furthermore they show that the kinase CKA2 is required for these responses.

 The results are very clearly described and presented however for some analyses (Fig2 B&C, Fig2 E&F, Fig3 A&B and in Fig4) wildtype and mutant lines are separated even though they come from the same experiment and could be directly compared and statistically significant differences noted so that the charts are more informative.

In the discussion, the statement "These immune responses are dependent on CK2-mediated phosphorylation of PFLP" is made. This seems highly likely but since the authors do not present direct evidence for this phosphorylation the wording might be too strong.

Since the complementation line was under the 35S promoter it would be interesting to see CKA2 gene expression levels in wildtype and transgenic lines, (and the in response to PTI if relevant).

Minor points:

Fig2E Callose intensity axis label incomplete

296 cka2 italics

318 formatting error in flg22Pst

319 and the first instance on 407 CKA2 should be uppercase italics

Fig5 legend, tenses are inconsistent, 334 A) shows 336 C) indicates D) indicates

Author Response

The authors have made some interesting findings regarding the mechanism by which PFLP enhances PTI responses, showing that the key phosphorylation site is necessary for these responses but that in vivo phosphorylation is required. Furthermore they show that the kinase CKA2 is required for these responses.

The results are very clearly described and presented however for some analyses (Fig2 B&C, Fig2 E&F, Fig3 A&B and in Fig4) wildtype and mutant lines are separated even though they come from the same experiment and could be directly compared and statistically significant differences noted so that the charts are more informative.

Response: We sincerely thank the suggestion. In order to see the differences between mutants, transgenic line and wild-type plants, we did not make changes to the previous figures. Instead, we can see from Figures 5 and 6 that they should meet the reviewer’s comments.

In the discussion, the statement "These immune responses are dependent on CK2-mediated phosphorylation of PFLP" is made. This seems highly likely but since the authors do not present direct evidence for this phosphorylation the wording might be too strong.

Response: Thank you for your very pertinent suggestions. I agree with them very much and have adjusted my tone and wording based on your suggestions at the end of the discussion.

Since the complementation line was under the 35S promoter it would be interesting to see CKA2 gene expression levels in wildtype and transgenic lines, (and the in response to PTI if relevant).

Response: Thanks for your comment. The expression of CKA2 in CKA2R was detected and showed the results in supplementary result Figure 1.

Minor points:

Fig2E Callose intensity axis label incomplete

Response: Thank you. We revised it according to the suggestion.

296 cka2 italics

Response: Thank you. We revised it according to the suggestion.

318 formatting error in flg22Pst

Response: Thank you. We revised it according to the suggestion.

319 and the first instance on 407 CKA2 should be uppercase italics

Response: Thank you. We revised all CKA2R according to the suggestion.

Fig5 legend, tenses are inconsistent, 334 A) shows 336 C) indicates D) indicates

Response: Thank you. We revised it according to the suggestion.

Reviewer 2 Report

Comments and Suggestions for Authors

The manuscript can only be accepted after the text has been corrected.

Main remarks:

-The description of the methods is incomplete in many places. These should be supplemented with details.

-The results of sections 3.5 and 3.6 included the results of sections 3.2 and 3.3. therefore, these two sections should be omitted. This will make the manuscript shorter, which may not be enough to write a full article.

-The discussion section is more of a summary of the results than a comparison and analysis of the results with literature data. Therefore, this section should be rewritten.

Details:

Abstract:

line18-21: The sentence is unclear and contradicts the following sentence. (The degree of phosphorylation does not affect the plant responses, suggesting that phosphorylation plays an essential role?)

line 30: It should have been stated earlier in the text that Pectobacterium causes soft rot.

Introduction:

line 37-44: The concept of PTI is confusing not only here in the text but probably also in the literature. However, it is generally accepted, that the elicitors of the PTI are conserved bacterial molecules (microbe associated molecular patterns, MAMPs), which can be recognized by plant receptor-like proteins located in the plant cell membrane. Both pathogenic and non-pathogenic bacteria possess these molecules (e.g. flagellin, EF-Tu etc.). This explains why PTI can be activated by both of this type of bacteria. In addition, bacteria possess these molecules even before entering the plant, so even dead bacteria can induce PTI. Since no specific bacterial activity is required to trigger PTI, PTI is activated immediately after entering the plant. That's why PTI is called the first line of defense. Importantly, HR is not usually induced during PTI. Although, as is usual in nature, there are always exceptions (e.g. Pseudomonas tabaci flagellin can induce HR in tomato). After entering into plant tissue, the plant pathogenic bacteria sense the plant environment and produce the pathogenic factors such as harpin(s), proteins of Type III export system and effectors. In incompatible situations these factors can cause HR. I think, because only metabolically active pathogenic bacteria produce harpin after sensing the bacteria the plant environment, harpin cannot be considered an elicitor of PTI.

The introduction should include more information about the various functions of CK2 kinase, especially since it may be related to important regulators involved in the regulation of resistance responses (e.g. SA, jasmonic acid, ABA). And this should also be taken into account when evaluating the results.

line: 67-83: The description of the planned and completed works is too long. It should be shortened.

Materials and Methods

Line 86: The exact name of the Arabidopsis CK2 gene should be included.

Line 88-91: More details of producing CKA2R complementation line should be included (primers, PCR details, cloning procedure, promoter of the plasmid etc.).

Line 91-93: method of the verification of homozygosity should be included.

Line 102: the incubation period of IPTG is missing.

line: 107-108: the verification of the results of HrpN protein purification is missing.

line 108: there is no information about Flg22Pst. The product number and/or sequence of the peptide should be included.

line 118: the method of site-directed mutagenesis is missing.

line 130-132: the verification of the results of protein purification is missing.

line 136: the concentration of PFLP protein and HrpN in the mixture should be specified separately.

line141: what does the “three independent experiment” means? Was the experiment repeated three times with three plant generations at different times, or the experiment was repeated three times with the same plant generation. This information should be added to the text.

line 168: the origin of the bacteria (source and/or citation) should be included.

line 169: not only the OD600 value but also the corresponding bacterial concentration should be given.

line 183: what does the 0 time point means? Does this apply to plants before inoculation or this time point show the plants after inoculation? If this corresponds to the latter, then time should not be written as 0, because inoculation takes time.

line 184: the methods of RNA purification and cDNA synthesis should be included.

line 189: the identification number of the tubulin gene should be included.

line 191: the methods(s) of the verification of PCR product after qPCR is missing.

line 193: It is not clear what this sentence means: “At least five samples of each treatment were analyzed as repeats in this assay.”

Results

line 239: The values cannot be determined from the figure, so they would have to be entered somehow (47.4-fold).

Section 3.2 and 3.3. It is unclear how this system works. How the PFLP recombinant protein and the CK2 kinase interact. Does the PFLP protein enter plant cells or does the CK2 kinase localize to the surface of plant cells?

line: 285-287: it is missing that this occurs in Col-0 plants.

line 293: : it is missing that this occurs in Col-0 plants.

line: 318. flg22^Pst instead of flg22Pst

Discussion

line 370-379: The explanation is not convincing. A more detailed explanation and more examples (especially plant-based) would be needed.

If it is true that the CK2 kinase is the regulators of resistance responses regulated by SA, jasmonic acid and ABA, in the cka2 mutants this responses could also influence the results of the experiments. If this is the case, then this should be taken into account in the discussion.

Author Response

The manuscript can only be accepted after the text has been corrected.

Main remarks:

-The description of the methods is incomplete in many places. These should be supplemented with details.

Response: Thank you for your suggestion. We have made the Materials and Methods more complete in the revised manuscript based on your suggestion.

-The results of sections 3.5 and 3.6 included the results of sections 3.2 and 3.3. therefore, these two sections should be omitted. This will make the manuscript shorter, which may not be enough to write a full article.

Response: Thanks for your suggestion. The main objective of sections 3.5 (Figure 5) and 3.6 (Figure 6) is to compare the plant immune response and disease incidence in the cka2 mutant after complementation with cka2 in CKA2R. As a result, the wild-type data are not central to this analysis and have been omitted. We therefore present only the results for the cka2 mutant and its complementary line CKA2R. We also reorganized the effects of PFLP treatment on PTI response and disease resistance.

-The discussion section is more of a summary of the results than a comparison and analysis of the results with literature data. Therefore, this section should be rewritten.

Response: Thank you for your suggestion. We have rethought and rewritten the discussion paragraph. I hope this version can contain more thoughts.

Details:

Abstract:

line18-21: The sentence is unclear and contradicts the following sentence. (The degree of phosphorylation does not affect the plant responses, suggesting that phosphorylation plays an essential role?)

Response: Thanks for your comment. We rewrote the sentence to make the meaning clearly. (Line 18-22)

line 30: It should have been stated earlier in the text that Pectobacterium causes soft rot.

Response: Thank you. We described it in the revised sentence in line 20.

Introduction:

line 37-44: The concept of PTI is confusing not only here in the text but probably also in the literature. However, it is generally accepted, that the elicitors of the PTI are conserved bacterial molecules (microbe associated molecular patterns, MAMPs), which can be recognized by plant receptor-like proteins located in the plant cell membrane. Both pathogenic and non-pathogenic bacteria possess these molecules (e.g. flagellin, EF-Tu etc.). This explains why PTI can be activated by both of this type of bacteria. In addition, bacteria possess these molecules even before entering the plant, so even dead bacteria can induce PTI. Since no specific bacterial activity is required to trigger PTI, PTI is activated immediately after entering the plant. That's why PTI is called the first line of defense. Importantly, HR is not usually induced during PTI. Although, as is usual in nature, there are always exceptions (e.g. Pseudomonas tabaci flagellin can induce HR in tomato). After entering into plant tissue, the plant pathogenic bacteria sense the plant environment and produce the pathogenic factors such as harpin(s), proteins of Type III export system and effectors. In incompatible situations these factors can cause HR. I think, because only metabolically active pathogenic bacteria produce harpin after sensing the bacteria the plant environment, harpin cannot be considered an elicitor of PTI.

Response: Thank you for your clear and correct opinion. PTI often does not cause hypersensitive reactions, which depends on the intensity of the defense response induced. However, the plant pathogenic bacteria will use T3SS to secrete effector proteins to establish infection after approaching the plant cell. In this process, although harpin is related to T3SS, this protein will not enter the plant cells but can induce stronger defense signals and even cause HR. In this manuscript, we just attend to emphasize the phosphorylation of PFLP is required for its functional enhancement of PTI-mediated defense responses. In order not to overemphasize whether harpin is a type of PAMP, flg22 were used for research in subsequent experiments. According to your suggestion, we also rewrote this sentence.

The introduction should include more information about the various functions of CK2 kinase, especially since it may be related to important regulators involved in the regulation of resistance responses (e.g. SA, jasmonic acid, ABA). And this should also be taken into account when evaluating the results.

Response: Thank you for your reminder. We have added the introduction to the function of CK2 and added the references.

line: 67-83: The description of the planned and completed works is too long. It should be shortened.

Response: Thank you for your suggestion, we have made this paragraph more concise.

Materials and Methods

Line 86: The exact name of the Arabidopsis CK2 gene should be included.

Response: Thank you. The information was added in the revised version.

Line 88-91: More details of producing CKA2R complementation line should be included (primers, PCR details, cloning procedure, promoter of the plasmid etc.).

Response: Thank you for the comment. More detail information was written in the materials and methods (line 121-128). And, the sequences of primers and confirmed result were added in the supplementary results.

Line 91-93: method of the verification of homozygosity should be included.

Response: Thanks for the comment. The method for verification of homozygous lines were written in the revised manuscripts (line 128-136)

Line 102: the incubation period of IPTG is missing.

Response: Thank you for your suggestion, we have modified it according to your suggestion (line 145-147).

line: 107-108: the verification of the results of HrpN protein purification is missing.

Response: Thank you for your suggestion, we have added how to verified the HrpN protein in the materials and methods (line152-154).

line 108: there is no information about Flg22Pst. The product number and/or sequence of the peptide should be included.

Response: Thank you for the remind, we have added the sequence of peptide and its source (reference) in line155.

line 118: the method of site-directed mutagenesis is missing.

Response: Thank you for the remind, we have added the method in line180-181.

line 130-132: the verification of the results of protein purification is missing.

Response: Thank you for your reminder. The verification results have been placed in Supplementary Figure 2. They are also noted in the Materials and Methods.

line 136: the concentration of PFLP protein and HrpN in the mixture should be specified separately.

Response: Thank you. The sentence was revised in line 206.

line141: what does the “three independent experiment” means? Was the experiment repeated three times with three plant generations at different times, or the experiment was repeated three times with the same plant generation. This information should be added to the text.

Response: Thank you for your suggestion. This experiment was indeed conducted at least three times, with more than 30 plants tested in each treatment in each experiment. To avoid confusion, we have corrected the description.

line 168: the origin of the bacteria (source and/or citation) should be included.

Response: Thank you. We added the source of this bacterial strain in the revised manuscript.

line 169: not only the OD600 value but also the corresponding bacterial concentration should be given.

Response: Thank you. The information was added.

line 183: what does the 0 time point means? Does this apply to plants before inoculation or this time point show the plants after inoculation? If this corresponds to the latter, then time should not be written as 0, because inoculation takes time.

Response: Thank you for your suggestion. This method is indeed not clearly stated. 0 hr does refer to the treatment without injection and inoculation. It has been corrected in the manuscript.

line 184: the methods of RNA purification and cDNA synthesis should be included.

Response: Thank you for the comment. The RNA purification and cDNA synthesis was added in revised manuscript (line 258-262).

line 189: the identification number of the tubulin gene should be included.

Response: Thank you. The gene number was added.

line 191: the methods(s) of the verification of PCR product after qPCR is missing.

Response: Thanks for the comment. The melting curve analysis was added in the manuscript.

line 193: It is not clear what this sentence means: “At least five samples of each treatment were analyzed as repeats in this assay.”

Response: Thank you. The sentence was rewritten to make it clear.

Results

line 239: The values cannot be determined from the figure, so they would have to be entered somehow (47.4-fold).

Response: Thank you for this suggestion. We did not mark the relative values ​​on the graph to make the figure look clearer, but for the sake of faithful presentation, we wrote the relative values ​​in the results so that readers can understand the differences.

Section 3.2 and 3.3. It is unclear how this system works. How the PFLP recombinant protein and the CK2 kinase interact. Does the PFLP protein enter plant cells or does the CK2 kinase localize to the surface of plant cells?

Response: Thanks for the comment. Since the experiment was conducted using the infiltration method, the recombinant protein and flg22 were both outside the cells of plant tissue. Since CK2 is also an extracellular secretory protein, it can be speculated that CK2 phosphorylates PFLP to enable it to more effectively amplify the PTI defense response occurring on the cell surface. This part is presented in the discussion. We also made corrections in the discussion.

line: 285-287: it is missing that this occurs in Col-0 plants.

Response: Thanks for the reminder. Sorry for the misunderstanding. We have rewritten the sentence to make it clearer.

line 293: : it is missing that this occurs in Col-0 plants.

Response: Thank you for your reminder. We have revised the content according to your suggestion to make the meaning clearer.

line: 318. flg22^Pst instead of flg22Pst

Response: Thank you. We have revised it according to the suggestion.

Discussion

line 370-379: The explanation is not convincing. A more detailed explanation and more examples (especially plant-based) would be needed.

Response: Thanks for your comment. In most studies, the use of simulated phosphorylation proteins can achieve functional effects. According to my search, only a few studies have achieved results close to ours. In the discussion, we added a paper in Arabidopsis that proved that the actual phosphorylation of a protein may not be replaced by mimic phosphorylation to assist our speculation.

If it is true that the CK2 kinase is the regulators of resistance responses regulated by SA, jasmonic acid and ABA, in the cka2 mutants this responses could also influence the results of the experiments. If this is the case, then this should be taken into account in the discussion.

Response: Thank you. This is excellent advice. We have rewritten the last two paragraphs of our discussion to make our point clearer.

Reviewer 3 Report

Comments and Suggestions for Authors

This study aims to demonstrate the critical role of phosphorylation at the C-terminal casein kinase II (CK2) site of ferredoxin-like protein for its impact on PTI reaction in Arabidopsis during infection. The authors utilize recombinant PFLP and its mutants to investigate this objective and the background of the knock-out mutant ck2 and the corresponding complementation line. They also assessed the expression of a few PTI-related genes, FRK1, WRKY22, and 29, to confirm their observations. The main conclusion of this work confirms their premise that phosphorylation is involved in conferring the CK2 involvement in PFLP stimulation of PTI response in Arabidopsis.

The results obtained are reasonable and support the main conclusion. My main question is why the authors have chosen to use recombinant proteins. Why not generate the mutant and overexpression lines and investigate everything in vivo? I think they need to make some justification for their approach, because exogenous application of proteins raises concerns about their penetration into the cells and the stability of proteins during uptake. They present some mock controls, but such studies still have to be taken with care.

Some minor comments:

  1. Please add the justification for selecting PTI-related genes. It is mentioned in the Discussion, but it would be helpful to mention it also in the results section before presenting the actual results.
  2. On lane 319, change from cka2, which means ko mutant, to CKA2, when talking about overexpression of the gene.
  3. When talking about the results of ROS and callose accumulation after treatment of mutant and complementation lines, I think to comparison between Col-0 and complementation plants has to be made, meaning does complementation of the cka2 just restore to the wild type plants, or is there some overcompensation.
  4. The SDS-Page gels for recombinant protein expression have to be shown to show how pure the preparations were.

Author Response

This study aims to demonstrate the critical role of phosphorylation at the C-terminal casein kinase II (CK2) site of ferredoxin-like protein for its impact on PTI reaction in Arabidopsis during infection. The authors utilize recombinant PFLP and its mutants to investigate this objective and the background of the knock-out mutant ck2 and the corresponding complementation line. They also assessed the expression of a few PTI-related genes, FRK1, WRKY22, and 29, to confirm their observations. The main conclusion of this work confirms their premise that phosphorylation is involved in conferring the CK2 involvement in PFLP stimulation of PTI response in Arabidopsis.

The results obtained are reasonable and support the main conclusion. My main question is why the authors have chosen to use recombinant proteins. Why not generate the mutant and overexpression lines and investigate everything in vivo? I think they need to make some justification for their approach, because exogenous application of proteins raises concerns about their penetration into the cells and the stability of proteins during uptake. They present some mock controls, but such studies still have to be taken with care.

Response: Thank you. We fully agree that in vivo approaches, such as using mutant and overexpression lines, can provide important insights. However, the decision to use recombinant proteins in this study was primarily based on experimental feasibility. Introducing transgenes into mutant or complemented lines would have introduced additional layers of complexity, particularly in terms of genetic background and time constraints. Therefore, we chose a more direct approach by applying recombinant PFLP proteins exogenously, which allowed us to focus on the functional relevance of the protein variants within a well-controlled system. To address concerns about protein uptake and stability, we included appropriate mock controls and try to use fresh extracted protein solutions to minimize protein degradation. Furthermore, previous studies have demonstrated that exogenously applied PFLP can support this approach in the investigations. As suggested, we also explain this in the Discussion. 

Some minor comments:

  1. Please add the justification for selecting PTI-related genes. It is mentioned in the Discussion, but it would be helpful to mention it also in the results section before presenting the actual results.

Response: Thank you. We have slightly enhanced the results.

  1. On lane 319, change from cka2, which means ko mutant, to CKA2, when talking about overexpression of the gene.

Response: Thank you. We revised it according to the suggestion.

  1. When talking about the results of ROS and callose accumulation after treatment of mutant and complementation lines, I think to comparison between Col-0 and complementation plants has to be made, meaning does complementation of the cka2 just restore to the wild type plants, or is there some overcompensation.

Response: Thank you for the comment. We also agree to make such a comparison. We have modified Figures 5 and 6 based on the suggestions.

  1. The SDS-Page gels for recombinant protein expression have to be shown to show how pure the preparations were.

Response: Thank you for the comment. We have included the data in Supplementary Figure 2 to confirm the material status. At the same time, we also added the related information in materials and methods.

Round 2

Reviewer 2 Report

Comments and Suggestions for Authors

A few minor corrections are needed to make the manuscript acceptable.

Abstract:

line 21: I still don't find the statement logical. Neither the non-phosphorylatable nor the phospho-mimetic mutant affected the defense responses. Nevertheless, it is written that phosphorylation may play an important role. Why? The text should be reworded to make it more understandable.

Materials and Methods:

line 123: pGMT vector? pGEMT? missing reference or manufacturer

line 125: pBI121 reference is missing

line 125: primers (Supplementary Table 2?)

line: Agrobacterium, Italic

line 146: the length of the incubation period of IPTG is missing.

line 244: the reference of the Pectobacterium carotovorum subsp. carotovorum strain 244 Ecc17 should be included.

line 268: the manufacturer of the real-time mix is missing.

line 271: identifier all of the Arabidopsis genes (including TUB2) should be included into the Supplementary Table 2.

Figure B,C, E,F: The figures should indicate whether they are Col-0 or cka2 plants, so they will be easier to interpret (as in Figure 3).

line 386: Tris buffer control.

The description of Figure 5 is incomplete. What does B, F, PF mean?

line 599: the word “occur” seems to be unnecessary

Author Response

A few minor corrections are needed to make the manuscript acceptable.

Response: Thank you very much for the detailed review, we agree with it and have made revisions to the manuscript based on the suggestions.

Abstract:

line 21: I still don't find the statement logical. Neither the non-phosphorylatable nor the phospho-mimetic mutant affected the defense responses. Nevertheless, it is written that phosphorylation may play an important role. Why? The text should be reworded to make it more understandable.

Response: Thank you for your good suggestions. After careful consideration, we have revised the way of presentation. We hope that such revision can bring more logical guidance.

Materials and Methods:

line 123: pGMT vector? pGEMT? missing reference or manufacturer

Response: Thanks for your suggestion. We have revised the materials and methods and provided the company information for the commericial product.

line 125: pBI121 reference is missing

Response: Thank you. We have provided the company information.

line 125: primers (Supplementary Table 2?)

Response: Thank you. The information was added in materials and methods.

line: Agrobacterium, Italic

Response: Thank you. We have revised it accordingly.

line 146: the length of the incubation period of IPTG is missing.

Response: Thank you. We have added the information in the materials and methods.

line 244: the reference of the Pectobacterium carotovorum subsp. carotovorum strain 244 Ecc17 should be included.

Response: Thank you. The reference was added in the manuscript.

line 268: the manufacturer of the real-time mix is missing.

Response: Thank you. We have provided the company information.

line 271: identifier all of the Arabidopsis genes (including TUB2) should be included into the Supplementary Table 2.

Response: Thank you. We have confirmed the information was list in the supplementary Table 2.

Figure B,C, E,F: The figures should indicate whether they are Col-0 or cka2 plants, so they will be easier to interpret (as in Figure 3).

Response: Thank you for your good suggestion. We have corrected the Figure 2, and hope it will be easier to get the message.

line 386: Tris buffer control.

Response: Thank you. We have revised it accordingly.

The description of Figure 5 is incomplete. What does B, F, PF mean?

Response: Thank you. We have added the information of these characters in the legend.

line 599: the word “occur” seems to be unnecessary

Response: Thank you. The word “occur” was deleted.

Reviewer 3 Report

Comments and Suggestions for Authors

Thank you for addressing my comments. I only have one additional request: add the image of SDS-PAGE gel stained by Coomassie or any other dye with your purified recombinant proteins in addition to westerns. Western blot demonstrates only that proteins are purified and in comparable amounts, but it does not show how pure the protein preparations were. Since you said it was quantified, it is important to know are your quantification was based on 100%, 50%, or less of the actual recombinant protein present in each fraction. 

Author Response

Thank you for addressing my comments. I only have one additional request: add the image of SDS-PAGE gel stained by Coomassie or any other dye with your purified recombinant proteins in addition to westerns. Western blot demonstrates only that proteins are purified and in comparable amounts, but it does not show how pure the protein preparations were. Since you said it was quantified, it is important to know are your quantification was based on 100%, 50%, or less of the actual recombinant protein present in each fraction. 

Response: Thank you for your suggestions. In order to let readers know the purity of the experimental materials, it is indeed necessary to provide this data. We have added the result of SDS-PAGE in Supplementary Figure 2. This data shows that most of the protein solution we used is the target recombinant protein.